



# Global vs local glacier modelling: a comparison in the Tien Shan

Lander VAN TRICHT[1*], Harry ZEKOLLARI[2,3,4], Matthias HUSS[2,3,5], Daniel FARINOTTI[2,3], Philippe HUYBRECHTS[1]

[1]Earth System Science & Departement Geografie, Vrije Universiteit Brussel, Brussels, Belgium
[2]Laboratory of Hydraulics, Hydrology and Glaciology (VAW), ETH Zürich, Zurich, Switzerland
[3]Swiss Federal Institute for Forest, Snow and Landscape Research (WSL), Birmensdorf, Switzerland
[4]Laboratoire de Glaciologie, Université libre de Bruxelles, Brussels, Belgium

[5]Department of Geosciences, University of Fribourg, Fribourg, Switzerland

*Corresponding author: Lander Van Tricht (lander.van.tricht@vub.be)

**Abstract.** Glaciers in the Tien Shan are vital for freshwater supply, emphasising the importance of modelling

their future evolution. While detailed 3D models are suitable for well-studied glaciers, regional and global

assessments rely on simplified approaches. However, their accuracy remains understudied. Here, we compare

the evolution of six glaciers in the Tien Shan using (i) a 3D higher-order ice flow model and (ii) a global glacier

model (GloGEMflow). Additionally, we explore the impact of using in-situ measurements of mass balance and

ice thickness, as opposed to relying on globally available data. Our findings reveal that the choice of mass balance

model complexity and calibration has a minimal impact on aggregated volume projections, with less than 3%

variation by 2050 and less than 1% thereafter. The use of a detailed versus a simplified ice flow model results in

some noticeable discrepancies in the first half of the century, with an 8% variation in aggregated volume change

by 2050. These disparities primarily stem from calibration, while the glacier evolution pattern remains

consistent, showing good agreement between the detailed and simplified model. In general, our results

demonstrate that the initial ice thickness estimation has the largest effect on the future remaining ice volume,

potentially resulting in 2 to 3, and even up to 4 times, more ice mass remaining. Our findings thus suggest that

when modelling small to medium-sized glaciers the emphasis should be on having a reliable reconstruction of

the glacier geometry rather than focusing on a detailed representation of ice flow and mass balance processes.




## 1. Introduction

Glaciers worldwide are shrinking because of climate change (Hock et al., 2019). With a 2°C increase in global air temperature compared to pre-industrial levels, about 28% of the global glacier volume is projected to be lost by the end of the century, while this figure increases to 33% and 41% with a 3°C and 4°C atmospheric warming, respectively (Rounce et al., 2023). This glacier volume loss significantly contributes to sea-level rise and strongly impacts water availability for human consumption, agriculture and hydropower (Huss and Hock, 2018; Farinotti et al., 2019b; Immerzeel et al., 2020; Yao et al., 2022). In Central Asia, the relative mass losses are expected to exceed the global mean, with up to 60-70% of all glacier ice likely to disappear by the end of the 21[st] century under a moderate warming scenario (Marzeion et al., 2020; Compagno et al., 2022; Rounce et al., 2023). Owing to the vital importance of glaciers in this region, this will threaten the water supply of mountain communities and millions of people living in the downstream dry lowland areas (Immerzeel et al., 2020; Leng et al., 2023; Shannon et al., 2023).

The importance of glaciers and their evolution has led to various recent research efforts to develop large-scale models that can simulate glacier evolution of all glaciers regionally or globally (Radic et al., 2014; Huss and Hock, 2015; Kraaijenbrink et al., 2017; Maussion et al., 2019; Zekollari et al., 2019; Rounce et al., 2023). The benefits of these approaches are considerable, allowing modelling the global glacier evolution, and providing unprecedented insights in the dynamics and evolution of remote and inaccessible glaciers. However, significant simplifications and parameterisations are utilised in these models, which could potentially affect the reliability of the glacier projections and related impacts. Therefore, detailed glacier evolution studies on individual glaciers are essential to evaluate the importance of simplifications used in global-scale models (Zekollari et al., 2022). A pilot study on this topic was conducted by Huss et al. (2010) who, when presenting the retreat parameterisation that forms the basis for glacier evolution in GloGEM (Huss and Hock, 2015), compared this simplified glacier evolution representation with simulations performed with a 3D full-Stokes model for two Swiss glaciers. The study revealed a good agreement in terms of glacier volume evolution but did not explore the model's sensitivity to different mass balance forcings or driving input data.

The objective of this study is to expand the sample of local, glacier-specific vs global-scale glacier evolution analyses by conducting a comparative analysis of the glacier evolution of six well-studied glaciers located in the Tien Shan, Central Asia. To achieve this, we employ two glacier evolution models: (i) a 3D higher-order ice flow model and (ii) the global-scale GloGEMflow model. Additionally, we investigate the impact of using either mass balance for each individual glacier based on direct observations, as compared to simplified, regional mass balance estimates. Finally, we also analyse the effect of employing ice thickness datasets that exhibit regional coverage (e.g., Farinotti et al., 2019a; Millan et al., 2022), in comparison to utilizing more intricate ice thickness distributions that have been reconstructed through in-situ measurements. The general workflow of this study is illustrated in Figure 1.



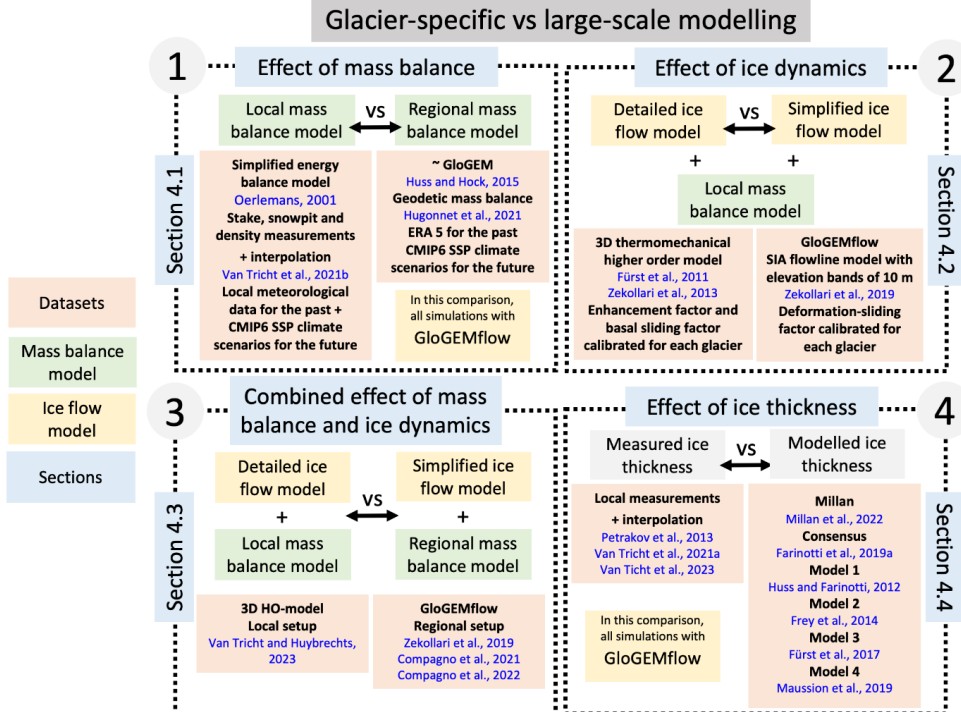

**Figure 1:** Study overview. The different blocks (numbered 1-4) show the four analyses performed in this study. The colours depict different conceptual elements of the study (legend given at left). For the individual datasets, key references are given (blue text).

## 2. Study area

In the analysis, we focus on six well-studied glaciers (Figure 2), distributed across four subregions of the Kyrgyz Tien Shan: Ashu-Tor glacier and Grigoriev ice cap (southern Terskey Ala-Too, Inner Tien Shan), Bordu glacier and Sary-Tor glacier (Ak-Shyirak massive, Inner Tien Shan), Kara-Batkak glacier (northern Terskey Ala-Too, Inner Tien Shan), and Golubin glacier (Kyrgyz Ala-Too, Ala Archa, north-western Tien Shan). The selected glaciers range in size between 2-7 km² and have a volume varying between 0.1 and 0.4 km³, qualifying them as small- to medium-sized glaciers in the region.

The Tien Shan is characterised by a continental climate with low precipitation, particularly in the inner and eastern areas (around 300-400 mm at 3000 m asl), except for the western and northern Tien Shan, which receive slightly more precipitation (ca. 600-700 mm at 3000 m asl) (Aizen et al., 1995). In winter, a branch of the Azores High extends over the Tien Shan, resulting in limited snowfall. The primary precipitation in the region occurs during the spring and summer seasons.



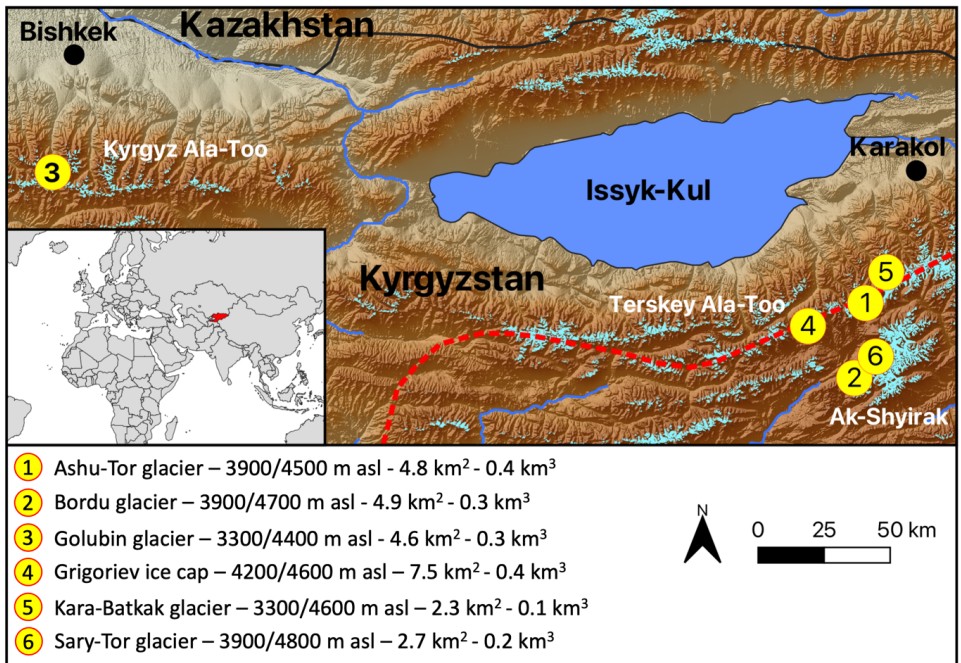

**Figure 2:** Location of the selected glaciers in the Kyrgyz Tien Shan. The background DEM is the SRTM (Jarvis et al., 2008), with light brown colours indicating low elevations and dark brown colours high elevations. Glaciers, as outlined in the Randolph Glacier Inventory version 6 (RGI consortium, 2017), are shown in light blue. Lakes and rivers are shown in a darker blue. The red dashed line shows the boundary between the western/northern Tien Shan (to the west and north of this line) and the inner/central Tien Shan, to the southeast of this line. different subranges of the Tien Shan, in which the selected glaciers are situated, are labelled in white. The small map shows the location of Kyrgyzstan. The six different glaciers are named below the map, accompanied with the elevation range, the area and the volume at time of the measurements.

Due to specific climatic conditions, the glaciers situated in the area experience substantial snow accumulation at higher elevations concurrent with primary ablation at lower elevations. This characteristic classifies the glaciers in the region as spring/summer-accumulation glaciers, as for example documented by Fujita (2008) and Van Tricht and Huybrechts (2022). This phenomenon renders the glaciers particularly vulnerable to the impacts of climate change. Rising temperatures not only contribute to increased melting but also significantly affect the type of precipitation (rain vs. snow), and thus total accumulation. All six selected glaciers (Figure 2) have been intensively investigated over multi-decadal time periods, including recent revisits and fieldwork (Hoelzle et al., 2017; Satylkanov, 2018; Van Tricht et al., 2021a; Van Tricht et al., 2021b; Van Tricht and Huybrechts, 2022; Van Tricht and Huybrechts, 2023).

For our analyses, we focus on two essential types of glacier data: (i) ice thickness, and (ii) mass balance. The ice thickness has been measured using Ground Penetrating Radar (GPR) on all six glaciers (Petrakov et al., 2014; Van Tricht et al., 2021a; Van Tricht et al., 2023). These measurements were used to reconstruct the ice thickness across the entire glacier area in Van Tricht et al. (2021a) and Petrakov et al. (2014), by relying on surface



characteristics such as slope and principles of ice dynamics, following methods described in Huss and Farinotti
(2012), Frey et al. (2014) and Fürst et al. (2017). Mass balance (MB) measurements have been performed since
the 1950s on Kara-Batkak glacier (Dyurgerov and Mikhalenko, 1995) and Golubin glacier (Azisov et al., 2022),
and since the 1980s on Sary-Tor glacier (Ushnurtsev, 1991) and Grigoriev ice cap (Mikhalenko, 1989; Fujita et
al., 2011). All MB programs were interrupted in the 1990s, but over the past decade, several MB programs have
been restarted (Hoelzle et al., 2017), including new measurements on Bordu glacier and Ashu-Tor glacier. Most
of these new measurements are conducted through various collaborations between (i) the Tien Shan High
Mountain Scientific Research Center as part of the CHARIS Project (Satylkanov, 2018), (ii) the Capacity Building
and Twinning for Climate Observing Systems (CATCOS) and Central Asia Water (CAWa) projects (Hoelzle et al.,
2017), and (iii) international partners from, for example, Switzerland and Belgium. The present-day mass balance
measurements include ablation stake, snow pit, and density measurements, which are used to estimate the
annual mean specific mass balance. Since a few years, several local automatic weather stations (AWS) have been
installed on and near the glaciers, providing detailed meteorological observations for mass balance modelling
(Hoelzle et al., 2017; Satylkanov, 2018; Van Tricht et al., 2021b).

## 3. Models and data

### 3.1. Mass balance models and climatic data

To identify the effect of variations in mass balance forcing on the glacier's future volume, we employ the global-
scale model GloGEMflow to simulate the glacier evolution, incorporating mass balance calculations generated
by (i) the mass balance component of GloGEMflow (regional mass balance model), and (ii) a simplified energy
balance model (SEB) (local mass balance model).

The local mass balance model relies on a SEB (Oerlemans, 2001) that has previously been calibrated and applied
to the selected glaciers (Van Tricht et al., 2021b). The model computes melt using a parametrised version of the
net energy balance and determines solid precipitation based on a temperature threshold. Input data for the
model is limited to hourly temperature and precipitation as well as a digital elevation model (DEM) to derive
insolation and shadowing. Model parameters are tuned to match modelled mass balance with mean specific
mass balance evaluations within elevation intervals of 100 m. The latter are derived through a combination of
stake observations, analysis of snow and firn depths, density assessments, and extrapolation (Van Tricht et al.,
2021b). Data from local weather stations are used to drive the local mass balance model over the historical
period. Specifically, three distinct timeseries are employed. The first time series, known as the Tien Shan –
Kumtor series, is used to drive the local mass balance model for the Ashu-Tor glacier, the Bordu glacier, the
Grigoriev ice cap, and the Sary-Tor glacier. The second time series, derived from observations at the Golubin
station located near the terminus of the glacier, is employed to drive the model for the Golubin glacier. Lastly,
the Chon-Kyzyl-Suu series is utilised to drive the model for the Kara-Batkak glacier. For more comprehensive
information regarding the three distinct time series, please refer to Van Tricht and Huybrechts (2023).



The regional mass balance model is the one of GloGEMflow, which aligns with the Global Glacier Evolution Model
(GloGEM) that utilises a positive-degree-day (PDD) methodology for computing melt in 10-meter elevation
intervals on a monthly basis (Huss and Hock, 2015). Snow accumulation is calculated based on an air
temperature threshold, and refreezing of rain and melt water within the snow and firn is estimated based on
heat conduction. The model is driven by monthly air temperature and precipitation according to the ERA-5 re-
analysis (Hersbach et al., 2020). Calibration of the model is carried out through a three-step procedure aimed at
matching the modelled mass balance with the geodetic glacier-specific mass balance between 2000 and 2019,
as provided by Hugonnet et al. (2021). The calibration procedure is performed by iteratively adjusting the
precipitation gradient, degree-day factors and air temperatures until a satisfactory fit ($\pm$ 0.01 m w.e. yr$^{-1}$) is
obtained. For further details refer to Huss and Hock (2015).

To model the future mass balance, both the local and the regional mass balance models are driven by
temperature and precipitation outputs from five different Global Circulation Models (GCMs) (BCC-CSM, CESM2,
EC-Earth3-CC, MPI-ESM1-2-HR, Nor-ESM2-MM) under Shared Socioeconomic Pathway (SSP) 2-4.5 – an
intermediate warming scenario. The five GCMs were selected to obtain a variety of models with diverse transient
climate response, ranging from a warmer variant (e.g., EC_EARTH3-CC) to a colder variant (e.g., Nor-ESM-MM)
(Hausfather et al., 2022). In order to align the output of the GCMs with observed climate data, a bias correction
process is applied. This involves adjusting the GCMs to match the mean and standard deviation of the
overlapping time period. Both the local and the regional model use the delta-method (Beyer et al., 2020) to
perform this bias correction.

**3.2. Ice flow models**

In order to assess the impact of different approaches in representing ice dynamics, the future evolution of the
six glaciers is simulated using (i) a detailed ice flow model: a 3-dimensional higher-order thermomechanical ice-
flow model (3D HO-model) (Fürst et al., 2011; Zekollari et al., 2014) and (ii) a simplified ice flow model: the
global-scale glacier evolution model GloGEMflow (Huss and Hock, 2015; Zekollari et al., 2019). In both setups,
the models are coupled with data from the local mass balance model.

The detailed ice flow model (3D HO-model) has already been employed to various ice masses ranging from
mountain glaciers (Zekollari et al., 2014) to ice caps (Zekollari et al., 2017a,b) and an entire ice sheet (Fürst et
al., 2013, 2015). In a recent study (Van Tricht and Huybrechts, 2023), the model was calibrated and applied to
the six selected glaciers. In this model, longitudinal and transverse stress gradients are considered (Blatter, 1995;
Pattyn, 2003; Fürst, 2011; Zekollari et al., 2013), which contrast with models that only rely on a local stress
balance through the widely used shallow-ice approximation (SIA) (Hutter, 1983). Unlike Full-Stokes models that
incorporate all stresses (Réveillet et al., 2015; Jouvet and Huss, 2019), the 3D HO-model neglects vertical
resistive stresses by assuming cryostatic equilibrium (Pattyn, 2003). Moreover, only the horizontal velocity
components are solved, and the vertical velocities are determined based on the assumption of ice



incompressibility. The local ice flow model operates on a horizontal grid resolution of 25 m and considers 21 layers in the vertical. Calibration involves tuning the enhancement factor and the basal sliding parameter to minimise differences between the modelled and measured ice thickness and observed surface velocities after a

transient run that matches the present-day glacier state. Further details on the model can be found in Fürst et al. (2011).

The simplified ice flow model (GloGEMflow) (Zekollari et al., 2019) is an extended version of GloGEM (Huss and Hock, 2015) in which ice-dynamical processes are explicitly represented. GloGEMflow has been used to model

glacier evolution in several regions around the world, such as the European Alps (Zekollari et al., 2019; Compagno et al., 2021a), Scandinavia and Iceland (Compagno et al., 2021b), and High Mountain Asia (Compagno et al., 2022). In GloGEMflow, ice flow is explicitly incorporated using the SIA in a flowline-based approach, which is similar to the widely-used Open Global Glacier Model (OGGM; Maussion et al., 2019). Glacier evolution is calculated using the continuity equation for ice thickness, and the ice flow is calibrated through a deformation-

sliding factor that considers internal ice deformation, basal sliding, and other effects such as lateral drag. The 3D geometry of ice masses is simplified by subdividing each glacier into elevation bands of 10 m. Separate glacier branches and tributaries are not treated separately but integrated into a single flowline representation for each glacier. Glacier cross-sections are represented using trapezoids. In the case of the Grigoriev ice cap, the Randolph Glacier Inventory v6 (RGI, 2017) divides the glacier into five separated branches. The future evolution of each

branch is modelled independently, and the total volume of the ice cap is obtained by summing the five individual branches afterward. In GloGEMflow, a deformation-sliding factor is calibrated by transiently modelling the glaciers and by minimising the differences in glacier volume and area with respect to the reference glacier given by the consensus estimate of Farinotti et al. (2019a). For further information on GloGEMflow, please refer to Zekollari et al. (2019).


### 3.3. Ice thickness

When evaluating the modelled glacier evolution, we rely on the ice thickness distribution obtained from field measurements conducted on the six selected glaciers (Petrakov et al., 2014; Van Tricht et al., 2021a; Van Tricht

et al., 2023) (Table 1). However, we also conduct a separate analysis to examine the impact of regional datasets on the modelled glacier evolution. For this experiment, we run GloGEMflow using ice thickness distributions from four independent large-scale ice thickness datasets (Huss and Farinotti, 2012; Frey et al., 2014; Fürst et al., 2017; Maussion et al., 2019) and the consensus estimate derived from these model results (Farinotti et al., 2019a). In addition, we also use the recent ice thickness estimate by Millan et al. (2022). For more details on the

different ice thickness datasets, consult Table 1.



**Table 1:** Description of the different ice thickness datasets used in this study.

| Reference | Dataset name | Methodology |
|---|---|---|
| Petrakov et al., 2014<br>Van Tricht et al., 2021a<br>Van Tricht et al., 2023 | Measurements | The ice thickness distribution is obtained from local measurements using a GPR system and inter- and extrapolation (using mass conservation or shear stress). |
| Millan et al., 2022 | Millan | The ice thickness is inferred from observed surface velocities (from satellite data) using the SIA. |
| Farinotti et al., 2019a | Consensus | Composite ice thickness based on ensemble of models 1-4 |
| Huss and Farinotti (2012) | Model 1 | Mass conserving approach. The mass flux is determined and converted into ice thickness by prescribing a constitutive relation. |
| Frey et al., 2014 | Model 2 | Shear-stress based approach. The local ice thickness is inferred from an estimation of the basal shear stress using the SIA. |
| Maussion et al., 2019 | Model 3 | Mass conserving approach. The mass flux is determined and converted into ice thickness by prescribing a constitutive relation. |
| Fürst et al., 2017 | Model 4 | Minimisation approach. The ice thickness is inverted as a minimisation problem. |


## 4. Results

### 4.1. Effect of mass balance

Regardless of the mass balance model utilised, under the considered moderate warming scenario, five out of six
glaciers are projected to disappear entirely by the end of the 21st century. The exception is given by Kara-Batkak
glacier, which is projected to preserve 3-19% of its 2020 ice volume by the end of the century (as illustrated in
Figure 3 and Table 2), especially when the mass balance is calculated using the local mass balance model. The
distinguishing characteristic of the Kara-Batkak glacier is its substantial elevation range and its thin ice, which, in
combination, result in a relatively short response time, and thus a faster approach to equilibrium.


Concerning the evolution of the aggregated ice volume, there is a strong similarity in the overall decline when
comparing simulations employing the local and regional mass balance models (Figure 3). By the year 2050, the
projected aggregated ice volume from both models exhibits a mere 3% disparity compared to the volume in
2020 (29% remaining when using the local mass balance model, 32% with the regional mass balance model)
(Figure 3). This disparity diminishes to insignificance by 2075. Towards the end of the century, the disparity
becomes zero as nearly all ice dissipates, irrespective of the mass balance model utilised (Table 2).


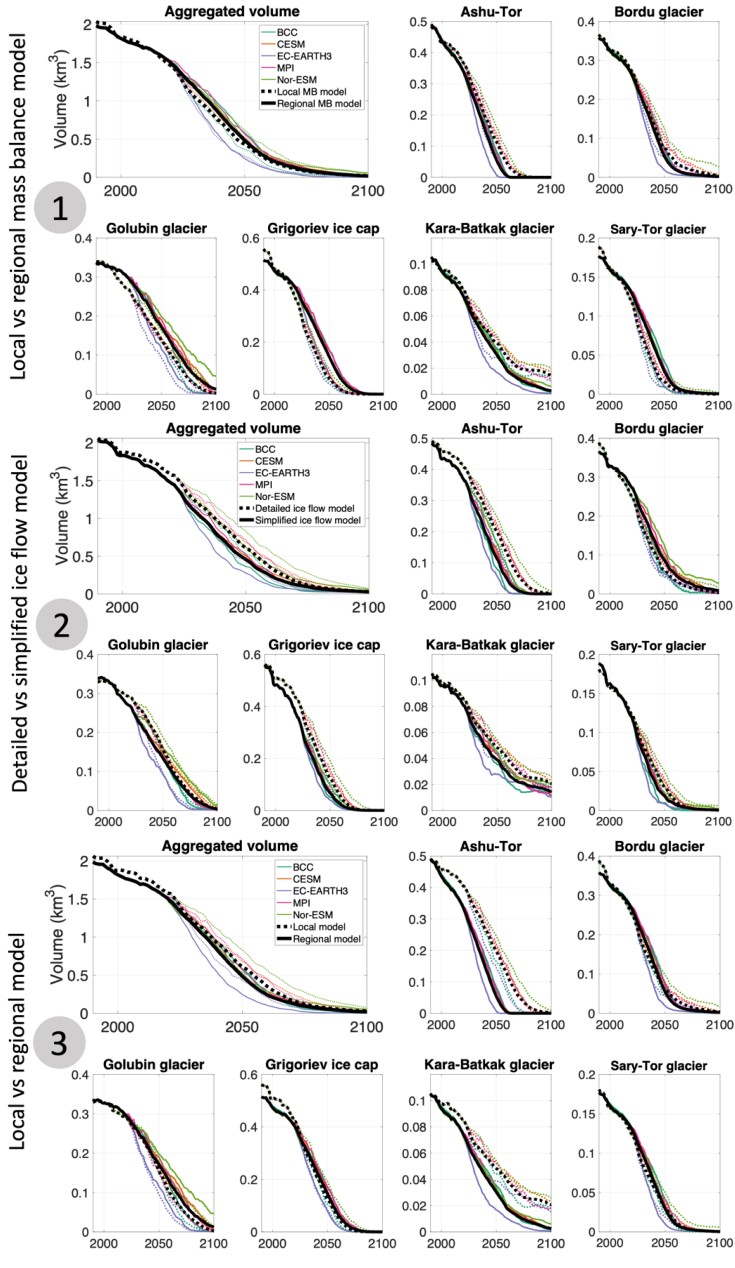

**Figure 3:** 1) Effect of mass balance model choice. Either the regional mass balance model (solid lines) or the local mass balance model (dashed lines) is used. For both cases, the ice dynamics are calculated with GloGEMflow. 2) Effect of ice flow model choice. Either the detailed ice flow model (dashed lines) or the simplified ice flow model (solid lines) is used. For both cases, the mass balance is calculated with the local model. 3) Combined effect of ice dynamics and mass balance model choice. Either the local model (mass balance and ice flow) or the regional model (mass balance and ice flow) is used. The line colours depict the five different GCMs (SSP2-4.5) while the thick black lines refer to the average of all GCMs. The future evolution of the aggregated ice volume is shown for the six considered glaciers and for their aggregated ice volume.

At the glacier-specific level, the observed discrepancies are somewhat more prominent, for instance for Golubin

glacier and Grigoriev ice cap (Figure 3). For the latter, using the local mass balance model results in a decrease

of 83% of the 2020 volume by 2050 while using the regional mass balance results in 64% of the 2020 ice volume

decrease (Table 2). This is also evident from Figure 4, which reveals that the Grigoriev ice cap retains more ice

when the regional mass balance model is employed. In the case of the Grigoriev ice cap, the mean mass balance

calculated by the local mass balance model is slightly lower than the geodetic value of 2000-2019 (Hugonnet et

al., 2021) (used for calibrating the regional mass balance model), with a deviation of about -0.08 m w.e. yr$^{-1}$.

Although this value may seem negligible, it could have significant implications for the future of the ice cap.

Indeed, Grigoriev ice cap has a limited altitudinal range, implying that even minor differences in the mass balance

can have a significant influence on its volume evolution. With respect to the Golubin glacier, the difference in

volume is primarily attributed to a ca. -0.8 m w.e. yr$^{-1}$ more negative mass balance in the local mass balance

model in the elevation bands between 3700 and 4000 m.

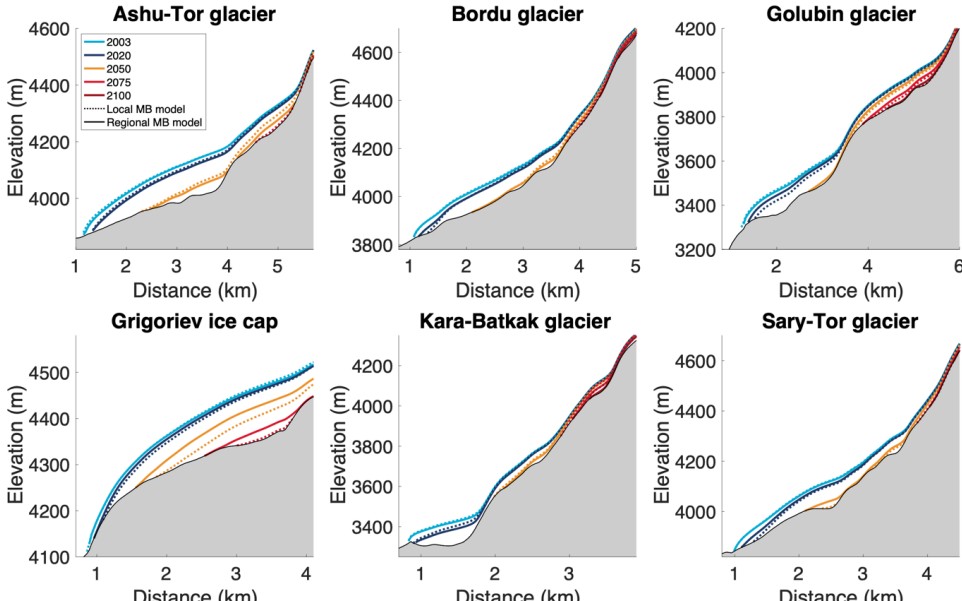

**Figure 4:** Two-dimensional glacier profiles simulated for various time intervals using the GloGEMflow model. The simulations
employ either the local mass balance model (represented by dashed lines) or the regional mass balance model (represented
by solid lines).

## 4.2.    Effect of ice dynamics

In the case of using different ice flow models but the same mass balance forcing, the course of glacier volume

decline between model runs is very similar (Figure 3). This suggests that for the selected medium-sized glaciers,



ice dynamics do not play a significant role in the future and that the volume evolution is mainly determined by the glaciers' mass balance.

The aggregated volume simulated throughout the 21st century by the detailed ice flow model is slightly larger
than that of the regional ice flow model (39% vs 31% in 2050 with respect to the 2020 volume), mainly due to time lag of the decline of the Grigoriev ice cap and the Ashu-Tor glacier in the beginning of the century (Figure 3 and Table 2). This translates into slightly more ice mass remaining by 2050 (28% using the detailed model and 19% using the simplified model for the Grigoriev ice cap, 50% vs 31% for the Ashu-Tor glacier, with respect to the 2020 volume) (Table 2). This difference is mainly attributed to the calibration method employed by each
model. Indeed, the regional model GloGEMflow minimises the discrepancy between the modelled volume at the inventory date (SRTM - bedrock) (Zekollari et al., 2019), while the local ice flow model (3D HO-model) minimises the difference between the modelled volume at the time of ice thickness measurements (2021 for the Grigoriev ice cap and 2019 for Ashu-Tor glacier) (Van Tricht and Huybrechts, 2023). In the second half of the century, the difference between both setups diminishes.


### 4.3.    Combined effect of mass balance and ice dynamics

In our next comparison, we vary both the ice flow model and the mass balance model, i.e. we compare the results of the local model to the regional model. While in some cases the differences are larger for individual
GCMs, for example for EC-EARTH3 and Nor-ESM, the multi-GCM mean glacier evolution of the two combinations is close to identical (Figure 3 and Table 2). The modelled glacier evolution is very similar also at the individual glacier level, except for the Ashu-Tor glacier for which we obtain a 50% reduction in ice volume with respect to 2020 for the local model and 85% using the regional model. The larger differences in future glacier evolution can be attributed to a mismatch between the local mass balance measurements used to calibrate the local mass
balance model and the geodetic mass balance used to calibrate the regional mass balance model, as well as the calibration procedure (see section 4.2). The mass balance derived for this glacier from the limited number of local measurements performed by the Tien Shan High Mountain Research Center (Satylkanov, 2018) may not be entirely representative (see Van Tricht and Huybrechts, 2023). On top of that, also the geodetically derived mass balance estimate for this glacier is notably lower than that for surrounding glaciers, indicating that the glacier
might be affected by peculiar local characteristics that are difficult to capture. Regarding Kara-Batkak glacier, the local setup retains more ice by the end of the century, particularly due to the larger amounts of accumulation predicted at the highest altitudes by the local mass balance model.






**Table 2:** Relative remaining ice volume compared to 2020.

| Local vs regional mass balance model. All simulations are performed with the simplified ice flow model | | | | | |
|---|---|---|---|---|---|
| Volume vs 2020 | 2050 | | 2075 | | 2100 |
| | Local model | Regional model | Local model | Regional model | Local model | Regional model |
| Ashu-Tor | 31% | 16% | 1% | 0% | 0% | 0% |
| Bordu | 28% | 20% | 7% | 3% | 2% | 1% |
| Golubin | 52% | 58% | 14% | 22% | 0% | 4% |
| Grigoriev | 17% | 36% | 0% | 2% | 0% | 0% |
| Kara-Batkak | 41% | 50% | 27% | 12% | 19% | 3% |
| Sary-Tor | 13% | 22% | 1% | 1% | 0% | 0% |
| Aggregated | 29% | 32% | 6% | 6% | 1% | 1% |
| Detailed vs simplified ice flow model. Both models use the local mass balance model. | | | | | |
| Volume vs 2020 | 2050 | | 2075 | | 2100 |
| | Detailed model | Simplified model | Detailed model | Simplified model | Detailed model | Simplified model |
| Ashu-Tor | 50% | 31% | 10% | 1% | 0% | 0% |
| Bordu | 17% | 33% | 6% | 9% | 1% | 3% |
| Golubin | 57% | 46% | 16% | 13% | 1% | 1% |
| Grigoriev | 28% | 19% | 0% | 0% | 0% | 0% |
| Kara-Batkak | 57% | 48% | 31% | 26% | 24% | 18% |
| Sary-Tor | 25% | 16% | 2% | 2% | 0% | 0% |
| Aggregated | 39% | 31% | 8% | 6% | 2% | 2% |
| Combined approach. Local vs regional model. | | | | | |
| Volume vs 2020 | 2050 | | 2075 | | 2100 |
| | Local model | Regional model | Local model | Regional model | Local model | Regional model |
| Ashu-Tor | 50% | 15% | 10% | 0% | 0% | 0% |
| Bordu | 24% | 20% | 10% | 3% | 1% | 1% |
| Golubin | 53% | 58% | 15% | 22% | 1% | 4% |
| Grigoriev | 29% | 36% | 1% | 2% | 0% | 0% |
| Kara-Batkak | 52% | 40% | 32% | 12% | 23% | 2% |
| Sary-Tor | 23% | 22% | 2% | 2% | 0% | 0% |
| Aggregated | 39% | 32% | 7% | 6% | 0% | 0% |

## 4.4. Effect of initial ice thickness

In addition to differences in model setup (sections 4.1 to 4.3), also input data variability can be an important factor when comparing local, glacier specific vs. global-scale modelling efforts. To analyse this effect, we model the future glacier evolution by varying the ice thickness reconstruction – arguably one of the most important datasets when representing glaciers. More specifically, we model the glacier evolution with seven different ice thickness distributions as initial state (see section 3.3. and Table 1 for details) using the simplified ice flow model and the regional mass balance model.



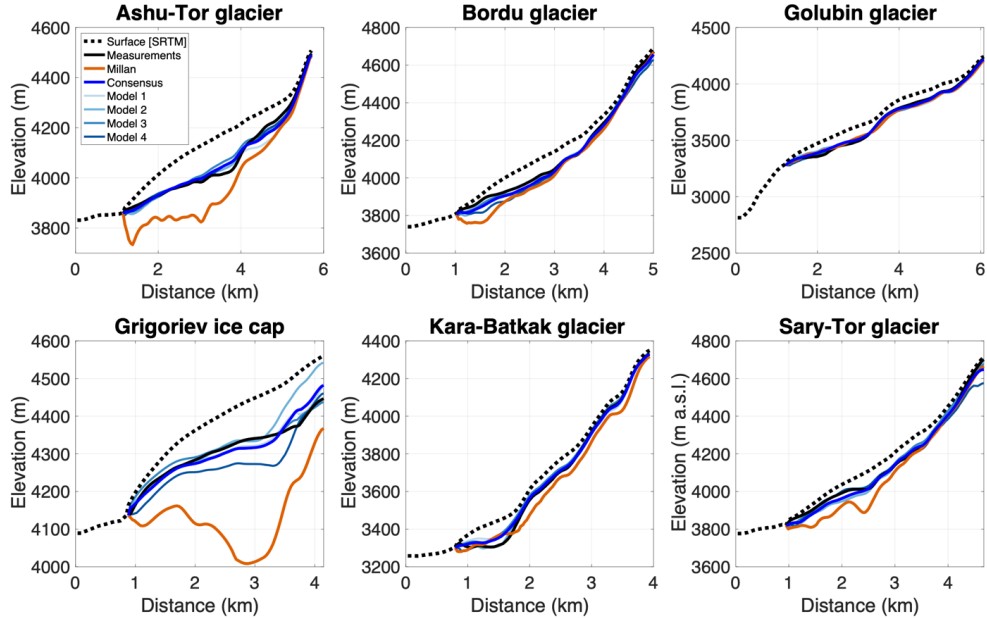

**Figure 5:** Surface and bedrock elevation from the different glacier geometry reconstructions used in the analysis. For the Grigoriev ice cap, the bedrock and surface elevation are shown for the largest branch (RGI13.06741). Models 1-4 were used to compile the consensus estimate of Farinotti et al. (2019a) (blue) and are shown as thinner lines. In all cases, the surface elevation corresponds to the Shuttle Radar Topography Mission (SRTM) (Jarvis et al., 2008). For more information on the different ice thickness datasets, refer to Table 1.

In general, the ice thickness reconstructions from the four models (referred to as model 1-4) used to generate the consensus estimate exhibit close agreement with both each other and the ice thickness measured in the field (Figure 5), displaying variances in the range of metres to tens of metres. In contrast, the more recent ice thickness estimate by Millan et al. (2022) generally suggests larger ice thicknesses (Figure 5). This is particularly notable for the Grigoriev ice cap and the Ashu-Tor glacier, which are characterised by a very limited surface motion (Van Tricht et al., 2023). For example, in the case of the Grigoriev ice cap, the dataset by Millan includes ice thicknesses of up to 400 m in the central region, which surpass the measured ice thickness by a factor of three. For Golubin glacier, all the available ice thickness datasets are very similar, with an almost complete overlap between reconstructed bedrock topography (Figure 5).


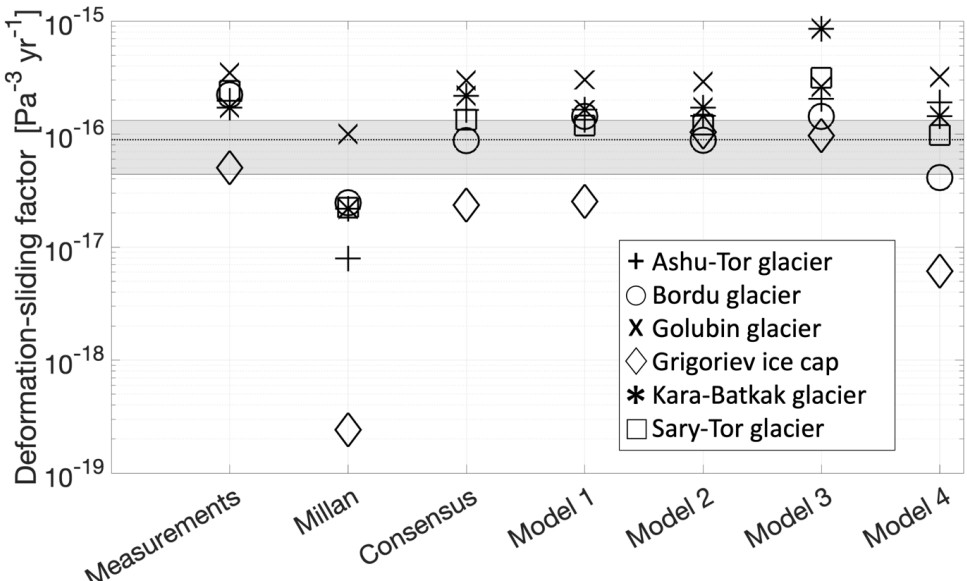

**Figure 6:** Calibrated deformation-sliding factor for every individual ice thickness dataset. The dashed horizontal black line shows the mean value reported in the literature study of Zekollari et al. (2022). The grey interval corresponds to the lower and upper quartiles. The different ice thickness datasets are described in Table 1.


The comparison is performed with the simplified ice flow model GloGEMflow, which requires a specific

calibration of the deformation-sliding factor for each ice thickness datasets to match the ice thickness at inventory date. The calibrated deformation-sliding factors are all found to be in the same order of magnitude (approximately 1-2 x 10$^{-16}$; Figure 6), except for the dataset by Millan et al. (2022). In this latter case, the deformation-sliding factor required to match the surface geometry at inventory date is about one order of magnitude lower (see again Figure 6).


The lower deformation-sliding factors required when using the Millan dataset are a result of the much larger ice thickness that is reconstructed by that dataset compared to others (Figure 5). Indeed, given the Millan ice thickness, the surface geometry at inventory date can only be matched by assuming much stiffer ice and thus much slower ice flow. This is particularly evident for the Ashu-Tor glacier and the Grigoriev ice cap, where

calibrated deformation-sliding factors of $7.9 \times 10^{-18}$ Pa$^{-3}$ yr$^{-1}$ and $2.4 \times 10^{-19}$ Pa$^{-3}$ yr$^{-1}$ are found, respectively. The deformation-sliding factors for the consensus estimate (Farinotti et al., 2019a) and the thickness based on measurements are more consistent with literature values, albeit at the higher end. The factors resulting from the Millan dataset are instead only partially in agreement with literature values, and at the very low end of the spectrum (Zekollari et al. (2022) reported a mean value of $0.89 \times 10^{-16}$ Pa$^{-3}$ yr$^{-1}$ based on 48 published studies).




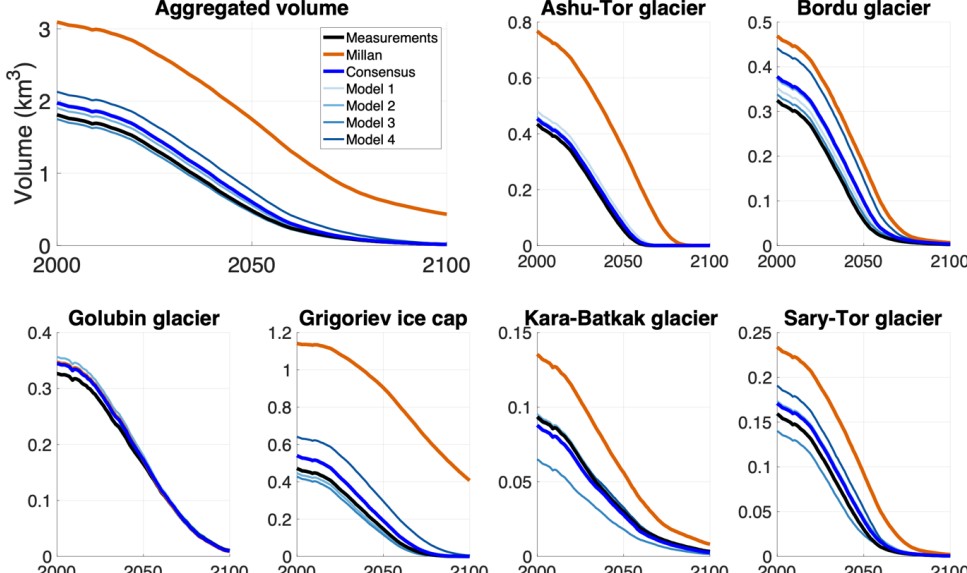

**Figure 7:** Future evolution of the glacier volume when relying on different ice thickness datasets (Measurements, Millan, Consensus, and Model 1-4) as initial state. All simulations are performed using GloGEMflow and represent the average of the five GCMs under SSP2-4.5. The black line shows the result when the simulations are performed by using the ice thickness dataset which is based on in-situ measurements. The different ice thickness datasets are described in **Table 1**.

Following the calibration for each individual ice thickness reconstruction, the future evolution of the glaciers is modelled. The findings reveal that the initial ice thickness (volume) reconstruction strongly dictates the glacier's future evolution, with larger initial volumes resulting in more ice remaining by 2050 and the end of the 21st century (Figure 7). The discrepancies between the various configurations diminish throughout the 21st century (Figure 7 and Figure 8), which is an expression of the differences in ice thickness being particularly pronounced in the lower parts of the glacier (Figure 5).



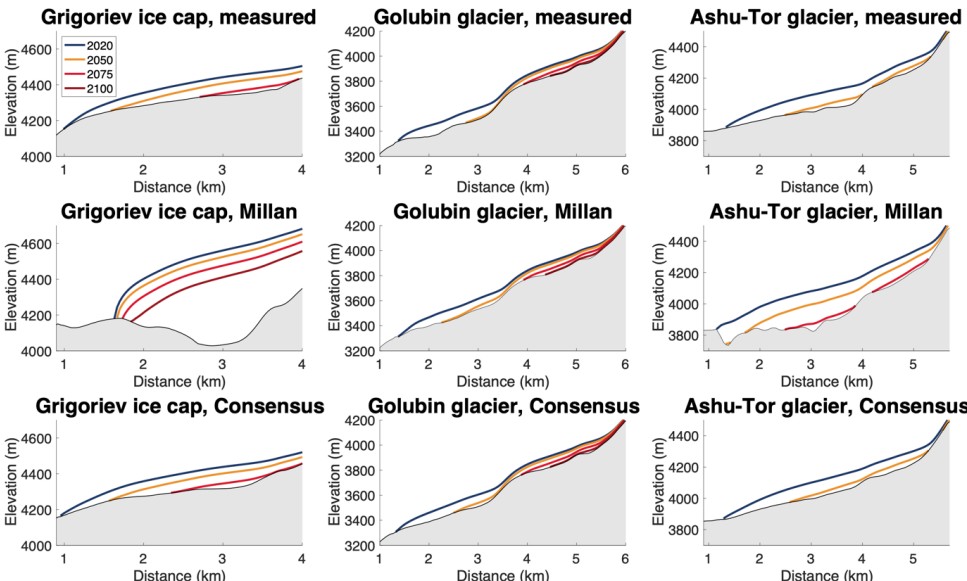

**Figure 8:** Simulated two-dimensional glacier profiles for three glaciers (columns) at different time steps (colours) are generated using the GloGEMflow model. The simulations are initialised using three distinct ice thickness distributions (represented by rows).

The Grigoriev ice cap exhibits the most substantial contrast in simulated ice volume when comparing the different datasets. In 2050, modelling using the Millan et al. (2022) dataset indicates an ice volume of 0.89 km$^3$, whereas this value only is between 0.20 km$^3$ and 0.17 km$^3$ when considering the consensus estimate and the ice thickness measurements, respectively, as initial state (Figure 7). Conversely, the differences in ice volume for the Golubin glacier between the middle and end of the century are negligible, as the various ice thickness datasets demonstrate a close resemblance (see Figure 5).

In the case of the Ashu-Tor glacier, the simulated ice volume is relatively similar across different datasets when starting from measurements, the consensus estimate, or models 1-4, with approximately 0.07 km$^3$ at the middle of the century (Figure 7). However, when utilising the Millan dataset as a starting point, the ice volume at the middle of the century is found to be 0.35 km$^3$. As all ice ultimately disappears, the results from all datasets converge by the end of the century (Figure 7). For the other three glaciers, the disparities in remaining ice mass throughout the century are rather small, and they diminish further towards the end of the century due to substantial ice loss or smaller variations in bedrock elevations in the accumulation area.



## 5. Discussion

The comparison between the two different mass balance approaches showed little influence on future glacier evolution, as demonstrated by the close agreement between the simulations based on the local mass balance
and the regional mass balance forcing in Figure 3 and Figure 10. While the local mass balance model has been specifically calibrated to match the data shown in Figure 9b, extrapolated point mass balances averaged over elevation bands of 100 m and averaged over 12 years, the regional mass balance model only uses the 20-year geodetic mass balance of Hugonnet et al. (2021) for calibration. This accentuates the striking resemblance between the modelled volume evolutions.

The mean extrapolated mass balance in 100 m elevation bands for Golubin glacier, which has the longest series of recent mass balance measurements from the selected glaciers (from 2010 to present-day) (Azisov et al., 2022), further confirms this. Indeed, a strong agreement is found between the mass balance calculated with the local mass balance model and the regional mass balance model (Figure 9b). The similarity is strong for most of
the other glaciers too. Only in the upper bands of the accumulation area, some mismatch occurs between the mass balance models, which we attribute to the estimated decline in accumulation in the upper reaches due to wind erosion and reduced moisture content in the provided mass balance data (WGMS, 2021). These processes are not directly taken into account in the regional model. In contrast, the local mass balance model, which is calibrated based on this data, shows a close agreement also at the uppermost elevations (Figure 9b). However,
these uppermost elevations pose the greatest uncertainty in mass balance evaluations due to limited accessibility, necessitating extrapolation procedures.

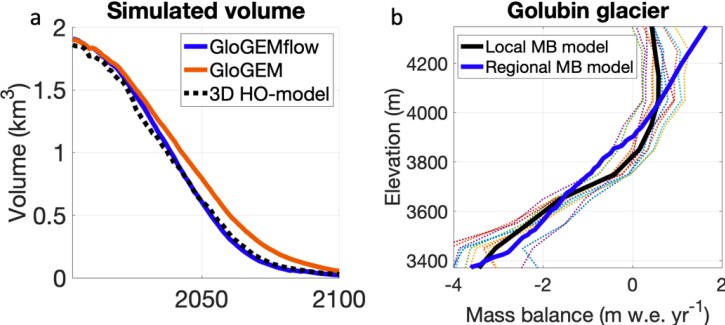

**Figure 9:** (a) Aggregated volume of the six considered glaciers as modelled with GloGEMflow (blue), GloGEM (orange), and
the 3D HO model (black). The results refer to the mean of five GCMs run under SSP2-4.5. (b) Modelled mean (2011-2021) mass balance of Golubin glacier in elevation bands of 100 m for the local and regional mass balance model. The dashed lines are the data from the WGMS for the 10 years being examined. This data is derived from measurements and extrapolation techniques.

The overestimated accumulation for the highest areas in the regional mass balance model (Figure 9b) could mean that the remaining future volume might be overestimated in these areas, with potential impacts on





regional-scale projections. That said, it should be noted that such high elevations are typically characterised by limited ice volume due to their relatively small area and high surface slopes. Furthermore, for certain specific glaciers, such as the Kara-Batkak glacier, the local mass balance model exhibits greater accumulation compared to the regional mass balance model. As a consequence, a larger quantity of ice is projected to persist at these higher elevations by the end of the century (Figure 3). In general, our findings suggest that the volume evolution aggregated over various glaciers is not substantially affected by differences between the two mass balance forcings. This is an encouraging finding because it supports the application of models like GloGEM, which use geodetic mass balances as the primary calibration mechanism. In contrast, local mass models such as the SEB model demand a greater amount of data, which is frequently lacking, particularly at regional scales.

Our analysis revealed that the representation of ice dynamical processes (3D HO-model vs flowline model) has a very limited effect on the modelled future glacier evolution (Figure 3 and Figure 10). The calibration procedure was found to have a more significant influence on the outcomes compared to the actual representation of ice dynamics and thus requires careful consideration. Nevertheless, when comparing the simulations to the results of the empirically-derived glacier-elevation change curve (the so-called Δh parameterisation, Huss et al., 2010) used as a standard in GloGEM (Huss and Hock, 2015), we observe slightly larger differences (Figure 9a). Specifically, the mass losses predicted by GloGEM are somewhat less pronounced, especially during the mid-21st century. We hypothesise that this may be due to the limited elevation range of the six glaciers we analysed, which results in the Δh parameterisation removing proportionally less ice at middle and high elevations (Zekollari et al., 2019). By the end of the century, the differences between GloGEM and the dynamical approaches (GloGEMflow, 3D HO model) are negligible, as nearly all ice disappears.

Generally, we found the differences in our simulations to be larger between individual GCMs than between individual model setups (Figure 3). This confirms that the major source of uncertainty when predicting the future evolution of the considered glaciers for a particular initial ice thickness is not glacier model complexity, but the choice of the climate model. This result is in line with the study by Marzeion et al. (2020), who partitioned the uncertainty in model-derived projections of global glacier mass change.

One of the key outcomes of this study is the pronounced sensitivity of the projected future glacier volume to the selection of the ice thickness reconstruction employed for model initialisation (Figure 10). This sensitivity results in substantial disparities in the modelled absolute glacier changes, as well as notable variations in the relative glacier changes when comparing different ice thickness reconstructions. The differences are particularly pronounced in the first decades of the simulation (Figure 7) and tend to decrease during the 21st century. Hence, the selection of a specific initial ice thickness has the potential to significantly influence projections of overall glacier runoff, a finding that aligns with the conclusions drawn by Huss et al. (2014). Furthermore, our analysis also revealed that the consensus estimate aligns well with field measurements, suggesting that the product might be well suited for regional glacier studies in Central Asia. In contrast, the Millan datasets generally





produces higher ice thicknesses, an effect that is particularly pronounced for slow-flowing glaciers such as Ashu-Tor glacier and the Grigoriev ice cap.

It should be emphasised that our glacier sample consists of small- to medium-sized glaciers only, as the largest glaciers in the region lack mass balance and ice thickness measurements. In future works, it could also be relevant to compare the outcomes of a complex ice flow -mass balance model with simpler approaches for larger

glaciers, such as the Enilchek glacier when focusing on the Tien Shan. Ideally, such analyses would be conducted in combination with fieldwork activities, to acquire ground-truth data for model input and calibration. Ice dynamics play a more significant role in the evolution of such large glaciers, and simplified flow models may yield more substantial differences compared to more complex ice flow modelling approaches. The agreement between the modelled volume evolution for the different model setups for our selected six glaciers is

encouraging nevertheless, and highlights that a global-scale flowline model is capable of accurately simulating glacier dynamics and evolution.

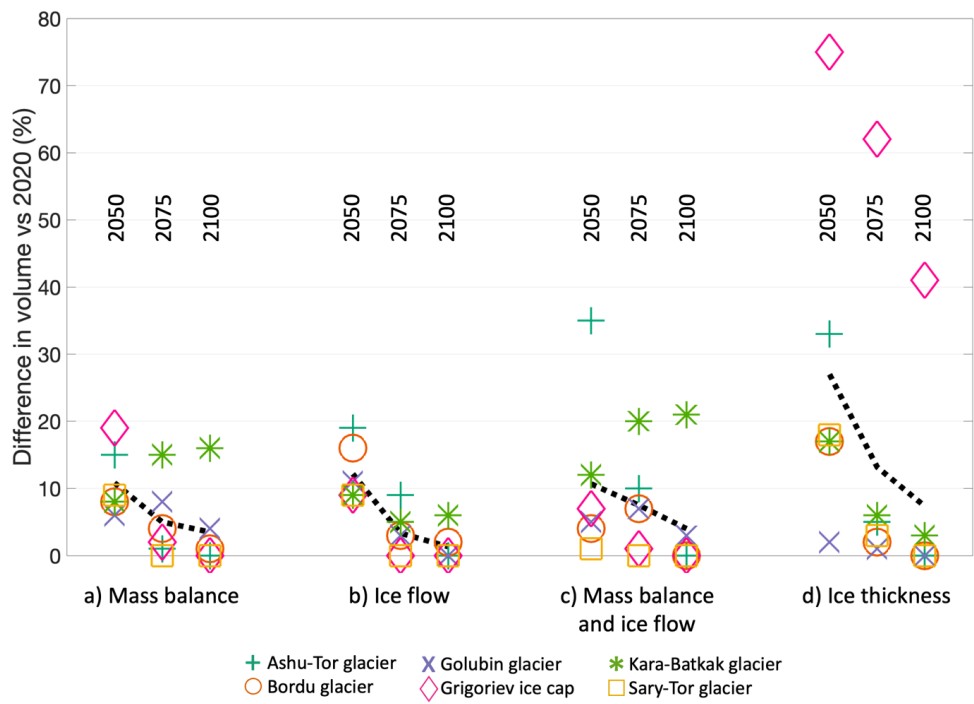

**Figure 10:** Relative difference in modelled ice volume for the six considered glaciers in three selected years throughout the

21st century. Results are shown for a) local mass balance model vs regional mass balance model, b) detailed vs simplified ice flow model, c) local mass balance model and detailed ice flow model vs regional mass balance model and simplified ice flow model, and d) different initial ice thickness distributions (seven different reconstructions; see Sect. 3.3 and **Table 1** for details) with respect to the ice volume in 2020 based on the measurements. The dashed black lines represent the evolution of the mean of all glaciers.



## 6. Conclusions

Regional to global scale glacier models are designed to simulate the evolution of a large number of glaciers. In this study, we assessed the accuracy and applicability of the GloGEMflow global glacier model and regional datasets for calculating changes in glacier volume. Specifically, we compared simulations from a local model (3D higher-order model coupled with a simplified energy balance model) with results from GloGEMflow for six glaciers in the Tien Shan.

Our simulations project a marked retreat of the glaciers for both the 3D higher-order model and GloGEMflow, with nearly all ice mass disappearing by the end of the century regardless of the complexity of the ice flow and mass balance model component. We found very similar results across our simulations, independently of the considered combination of ice flow and mass balance representation, notably ranging from simplified regional approaches to complex approaches that require glacier-specific field observations. Our analysis reveals that differences in the simulations are mostly driven by differences in the present-day ice thickness distribution, with up to four times more ice remaining at mid-century depending on the initial ice thickness, and in the climate evolution projected by a given GCM, rather than by differences in glacier model complexity. The influence of the initial ice thickness is especially large in the coming decades, as differences between thickness datasets are particularly pronounced in the lower part of the glaciers, which are first affected by glacier retreat.

Our study benchmarked the performance of a global-scale model when simulating small- to medium-sized glaciers, and follow-up studies might want to focus on similar comparisons for larger and more dynamic glaciers. We conclude that despite the simplifications that are necessary for global-scale glacier models, their performance is comparable to the one of more complex modelling frameworks driven by in-situ observations. Rather than focusing on complex model approaches, the focus should be on selecting adequate input data and reliable climate forcing, as these factors dominate the projected future glacier evolution.

## 7. Author contribution

The simulations with GloGEMflow were performed by LVT with direct help and assistance from HZ. MH modelled the mass balance using GloGEM. The 3D model runs were performed by LVT with the original code from PH. DF and PH provided supervision during the work. The manuscript and figures were created by LVT with help from all co-authors.

## 8. Competing interests

Some authors are members of the editorial board of "The Cryosphere". The peer-review process was guided by an independent editor, and the authors have also no other competing interests to declare.



## 9. Financial Support

LVT acknowledges a grant for a stay abroad by the Research Foundation-Flanders, which allowed him to perform a substantial part of this research during a research stay at ETH Zurich. PH, HZ, MH, and DF acknowledge the
funding received from the European Union's Horizon 2020 research and innovation programme under grant agreement No. 869304 (PROTECT).

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
