# Peer review of "Global vs local glacier modelling: a comparison in the Tien Shan"

_The Cryosphere, 2023_

## Referee Comment (RC1)

**General comments**

The study presents an intercomparison of volume projections for the 21$^{st}$ century for six glaciers in Tien Shan using multiple different modelling approaches, namely: two approaches in simulating surface mass balance, two approaches in simulating ice flow, and the use of different initial ice thickness estimates as input to the models.

I don't want to sound overly critical, but the impression I am getting (having some experience in supervising grad students over many years) is that the study was undertaken as a necessity to produce one more (often final) chapter for the doctoral thesis, and this project seemed like a relatively straightforward way to achieve this. Regardless if this is the case or not, this study needs to be a stand-alone contribution to scientific knowledge in order to be publishable. The current study fails to do so.

Firstly, the Introduction section is relatively short and poorly addresses some key elements expected to be part of this section, including: setting the context for the study, identifying knowledge gaps, and outlining the relevance and motivation for the objectives of the study. The lack of these elements does not only undermine the quality of presentation (which is relatively poor), but more importantly misses to highlight a novel research objective(s) or question(s) that authors aim to address. It took me a full read of the manuscript to realize that there is no novelty after all, and thus the omission of those elements from the Introduction section may be done on purpose.

All the models used in the study have been previously published and in most cases applied in many other studies. In fact, the leading author has published the results from the 3-D ice flow model applied on the same glaciers as in this study. This for itself does not mean that a novel analysis of previously published models cannot be done, as there are many publications that focused on model intercomparison such as in Farinotti et al., 2019, Hock et al., 2019, Marzeion et al., 2020, Edwards et al., 2021, to name just a few. However, all of these studies had that needed element of novelty that this paper fails to present. Let me elaborate on this a bit more by looking into different aspects of the study:

a) From the aspect of 'sensitivity analysis': Based on the conclusions that the projections are more sensitive to the choice of initial ice thickness than to choice of mass balance and ice flow models, I can view this study in the light of sensitivity analysis. As such one would expect to see a carefully designed set of experiments that would assure robustness of the analysis and results. After reading the paper, however, I realized that this is not the case. While claiming that the models of different complexities were compared, the authors failed to demonstrate how different these models truly are. In fact, scratching a bit more under the surface and going back to the original references behind these models, it become clear that the two mass balance models, as well as the two ice flow models, are very similar in their setup and application. The two mass balance models are both empirical, both forced by temperature and precipitation (as the only climatic drivers from ERA5 and GCMs), both calibrated to mass balance observations available for these glaciers (though using different mass balance datasets), and both applied locally (per glacier) – the latter makes me also wonder why call only one of this models 'local'. It is highly likely that the two models would produce very similar reconstructions of past mass balance for each of these six glaciers, although the authors do not show this. Similarly goes with the two ice flows modes (I give more details on this in the specific comments). The only modelling component that indeed shows large differences among the dataset is the initial ice thickness data, the differences which in some cases exceed 50% in the initial volume. Thus it is

not in any way surprising nor unexpected that the volume projections are most sensitive to the choice of this thickness data, much more so than to the choice of the two mass balance and two ice flow models. For a robust sensitivity analysis (sensitivity of regional glacier volume projections to different modeling components, such as ice thickness, ice flow, and mass balance) I refer the authors to Clarke et al. 2015.

b) From the aspect of 'model intercomparison'. The study fails to motivate or provide any relevance to their choice of the models of mass balance and ice flow. Considering that the study uses published models, some of which have been published by the same lead author for the same set of glaciers (Van Tricht and Huybrechts, 2023) I got an impression that the selection of model is based on convenience. GloGEMflow, or its predecessor GloGEM, has been extensively used in model intercompariosn studies (e.g. Hock et al.2019, Marzeion et al., 2020) . Furthermore, a similar version of the 'simple SEB' model applied here has also been used as part of sensitivity tests in GloGEM model on global scale (comparing the volume projections; Huss and Hock, 2015). So it is not really clear what knowledge gaps (if any) the authors are trying to address with this intercomparison. More to the point, if this study is publishable, so would be any study that takes a set of published models and intercompares their projections on a selected set of glaciers. Considering the availability of glacier models, but even so the number of studied glaciers worldwide, this could easily yield hundreds of publications. I hope you get my point.

c) From the aspect of 'model evaluation': since the study is not performing any model evaluation as this would require a reference dataset (preferably observations) to which the model simulations are compared to, some of the conclusions (such as 'a global-scale flowline model is capable of accurately simulating glacier dynamics and evolution') are inappropriate and unjustified. There is a potential to perform the model evaluation (I give some specific comments on that), but this would require a different model setup and calibration, as well as an independent validation dataset. However, even if this model evaluation is done correctly for the selected six glaciers, the question remains on how representative is the model performance for a large suite of glaciers over the region. And this would be one of the potential knowledge gaps to address.

The specific comments below have been initially written with the idea that the novelty of the study will somehow rise to surface if substantial revisions are made, especially in the Introduction section. However, considering the 'ill-posed' nature of the analysis, outlined with my points above, I doubt the revisions to the text can do the job. I hope the authors are able to find a research question or questions that can truly address at least some aspects of the knowledge gaps in the field. The current study fails to identify those gaps and subsequently to address any. Some potential avenues to consider, in my opinion, would be: (1) model evaluation targeted for the entire region (e.g. using the geodetic mass balance observations from Hugonnet et al. 2021 as validation dataset, while calibrating the models with in-situ mass balance observations and data used originally in GloGEM model), (2) coupling  the model with hydrology, to derive projections of glacier contribution to streamflow – these may be more relevant for the region than just the glacier volume projections, (3) application of a robust sensitivity analysis, such as the one based on Bayesian inference (see for example Rounce et al, 2020).

**Specific comments**

Line 20: Not clear what mass balance models were considered.

Line 49-50: Please expand a bit on this as the Intro is relatively short and leaves an impression that authors are not familiar with many inter-comparison modeling studies that have been done to date. Please mention studies that inter-compared the models of surface mass balance (e.g. temperature-index versus surface energy balance), as well as intercomparison studies of ice flow models of different complexities. While there may be not many of these performed for global scale, there are numerous inter-comparison studies done for individual glaciers, as well as a suite of glaciers, worldwide. I could be listing the references here but I do believe the adequate literature search should be done by authors not by the referees.

Line 58-59: There are a few logical discrepancies here that would be good to fix:

- 'to expand the sample' -> to expand the sample from what reference sample? It would be necessary to provide some information on roughly how many glaciers have been studied in this or similar way (i.e. intercomparison of models). See my comment from above: more references are needed on this topic.

- six well-studies glaciers located in the Tien Shan -> motivation (ideally a short paragraph) is missing on why this particular region and why these six glaciers

Line 60-61: The motivation is also missing on why these two models. Please include a few sentences on both models and state their relevance.

Consider that stating the motivation is necessary to communicate the relevance of your study to a more general glaciological audience than those who are in one way or another already involved in global glacier evolution modelling.

Figure 1: The workflow as displayed in the figure is useful,  as an overview of the methods and analysis, but should go to the Methods section, not to the Intro section. Please expand and revise the Intro section to better explain: background knowledge (and knowledge gaps in the field), context, motivation and relevance of your study. Currently, all these key elements of Introduction in a research paper are poorly addressed. Addressing them adequately would help highlighting the novelty of this study. Currently it's not clear to me what the novelty (as advancement of the knowledge in the field) actually is or will be with this paper. To state it more philosophically: not every new analysis is novel, and not every novel result needs to come from a new analysis.

Figure 2: Rather than showing this large map of the region, it would be much more informative to show the topographic maps of the six glaciers.

Line 100-101: I am assuming this explains why these six glaciers are selected. It would be useful to state this when you talk about motivation for the study.

Paragraph staring with Line 105: This paragraph should go under Data section as it describes the data that will be used in the study.

Section 3, Line 125: Here would be good to summarize the steps of the analysis and refer to the flow chart figure 1.

Also, call this section Data and Methods, and then first describe the data (include the bits from the 'Study area') and then summarize the methods (Figure 1) and present then in more details the models.

Line 131: It seems that it would be more accurate to call this an empirical SEB approach as it does not really differ much from an enhanced degree-day modeling (key driver of melt is temperature with some calculated effect of insolation). A SEB model generally assesses (or uses as input) key contributors to melt energy, including net shortwave, net longwave radiation and turbulent heat fluxes. This is not what your SEB model does, and since it is dependent on calibration with observed mass balance it is indeed an empirical approach.

Line 140-145: Shouldn't it be more correct (and less wordy) to state here that the model is calibrated and run over the observational period available for each glacier separately?

Line 150-152: It is not clear here if these mass balance observations are from each of these six glaciers, or from glaciers in this region as a regional (subregional) assessment. Also, note that Huss and Hock, 2015 performed their calibration on a regional scale (matching each glacier's mass balance to be equal to the regional mass balance), which is probably not the case here, so you can't just refer for details to their paper. Please explain the key differences in the approach originally used in GloGEMflow and the one you use here.

Line 155-156: Effectively you have the same input to both the SEB and the degree day model, and considering that only temp is taken from GCM, both models (of melt) are responding to temperature changes only. Make sure to make this clear when you are introducing the models of mass balance: these are two empirical mass balance models, mainly relating melt to temperature. The range of 'complexity' level here is thus not really large.

Line 161: Please specify what variables are used for this bias corrections, what observed climate data (ERA5 or something else?), and what the overlapping time period is. Also, is the bias correction considered on monthly basis? The results of the bias correction are sensitive to the choice of time scale (monthly vs annual) as shown across multiple studies taking part in GlacierMIP (Marzeion et al., 2020).

Line 183-185: Is this calibration performed for each of the six glaciers separately? Considering that the results of this model for the six glaciers have been published, it would be good to state some key results here in terms of the model accuracy and uncertainty. This would be a better use of the space than talking about the model setup considering that the model has been run and results published in Van Tricht and Huybrechts, 2023)

Line 198: It would be useful to show how representative this representation of cross-section is to reality (considering that you have the GPR data for these six glaciers).

Section 3.3: What is the reference year the ice thickness data is assessed for, and how to you assure that this year the same for all the models?

Line 224-225: Before showing the results for the projections (also what is the initial year?), it would be useful to show the comparison of reconstructed mass balance with ERA5 for each glacier (for example from 1979 onwards). I assume there is some overlap between the ERA5 period and the calibration period, but regardless of this it would be useful to show these results.

Line 276: 'ice dynamics do not play a significant role in the future and...' : No, do not over-extrapolate the results. This suggest that the results between the two ice models are similar, and nothing more.

Line 284-289: 'The description of details on the model setup and calibration should be given in the Methods section, not in the results.

Line 298-301: Again, it would be useful to see the reconstructed glacier evolution over the ERA5 period. The difference between the models in this period will influence the differences in the projections.

Line 303-305: Make sure that in the Results section you only present the results w/o discussing them. There are many occasions in the results section where you go into discussion as well as speculation about the model dissimilarities and what drives them.

Table 2: It would be more interesting to show the year when glacier is projected to loose 50% of its current volume, 75% of current volume and 100% of its current volume.
Also, if you can present this in a figure rather than in a table it would be even better.

Line 316: It's not the model setup but the choice of model

Line 319: 'arguably one of the most important datasets when representing glaciers.' : Remove the sentence as it is speculative or back it up with references. Also, the motivation for this analysis should come much earlier (when the objectives are stated), not in the Results section.

Line 321: Is initial state represented for the year 2020? Not clear from the Methods section. Also, if 2020 is the initial year, what thickness data was used during the calibration period (presumably starting before 2020) of both the mass balance and ice flow models?

Figure 7: It would be interesting to see how the evolution of glacier area (rather than volume) looks like, considering that area initially (at 2000 or 2020) is more similar (ideally the same) among the models. The glacier area is also easier to measure than ice thickness so the RGI should provide you some good reference.

Line 407-408: 'uses the 20-year geodetic mass balance of Hugonnet et al. (2021) for calibration.' It is not clear is it only for these six glaciers or regionally. If only for these six glaciers, than it's not surprising that the results are similar for several reasons: a) the models are similar as they are both basically temperature-index models, b) the model calibration is similar as it is tuning the model parameters to match the observed mass balance (per glacier), c) the glaciological and geodetic mass balance should not be very dissimilar from each other over the overlapping periods -> something that you could (and probably should) even show. also Note that while you are calling the Huss and Hock (2015) model a regional mass balance model, it is essentially a local model too as it is applied locally to each glacier. to restate my comment from before: the original model was tuned to regional mass balance observations (so that each glacier in the region has the same mass balance as the regionally-averaged mass balance). You did not specify how the calibration was done here, but it the geodetic mass balance data you used is available for each of the six glaciers over the 20-yr period (as well as over the 5 and 10 yr periods).

Line 408: 'This accentuates the striking resemblance between the modelled volume evolutions.' : I disagree that that this is a surprising result considering the many similarities in the modeling approaches (see above).

Line 436-438: But these two mass balance forcings are essentially very similar. How novel is the finding that similar forcing will give similar projections in similarly calibrated empirical models?

Line 440: 'In contrast, local mass models such as the SEB model demand a greater amount of data.' : Please see my comments from before on this. The SEB model you use here can also be calibrated with geodetic mass balance data. In terms of input climate data it also uses only temp and precip as GloGEM model does. Effectively, you are stating here that as long as there are measurements of glacier mass balance (whether glaciological or geodetic) the empirical models of mass balance can be calibrated. This is not really an insight.

Line 443: Similarly to the mass balance model, the two ice flow models are rooted in the same physics. In fact, the flowline model is an improved version of the empirical approach used in Huss and Hock (2015) whose goal was to provide a good match with simulations from a 3-D ice flow model (see for example Huss, M., Jouvet, G., Farinotti, D., and Bauder, A.: Future high-mountain hydrology: a new parameterization of glacier retreat, Hydrol. Earth Syst. Sci., 2010). So effectively you are comparing an empirical model, originally developed to resemble a 3-D ice flow model simulations, with another 3-D ice flow model. How different these two models are really? Your results effectively show that they may be not different after all, but again we knew that already considering the history behind the models' development, which btw you did not elaborate on.

Line 448: 'we observe slightly larger differences...': This is expected as GloGEMflow is improved to better resemble the 3-D ice flow simulations than the dh-parameterization approach.

Line The conclusion is invalid as the analysis is not consistent: you used five GCMs while only two models of mass balance and two models of ice flow (which are arguably not that different either).

Line 456-457: The conclusion is invalid as the analysis is not consistent: you used five GCMs while only two models of mass balance and two models of ice flow (which are arguably not that different either).

Line 458-459: This comparison is invalid (see my comment from above) and also it is not clear what results are 'in line' with what. Your results are not in line with partitioning of uncertainties - and this is how the sentence reads.

Line 461-462: You show that substantially different initial ice volume (>50% difference) leads to substantially different volume projections. This can not be a key outcome as this is just, pardon me being blunt here, common sense. Also, the high sensitivity of projections to initial volume has been already shown in Huss et al (2014) as you point out.

Line 468: Seems to me that this may be the only 'new' result in the paper, but I don't know how much it differs from what already has been found in Farinotti et al 2019. After all, the consensus estimates were proposed on the basis of the model performance relative to the ice thickness measurements.

Line 480-481: This is not what you were testing here. You did not do any evaluation of model performance but the intercomparison of future projections.

---

## Referee Comment (RC2)

**Global vs local glacier modelling: a comparison in the Tien Shan**

Lander VAN TRICHT, Harry ZEKOLLARI, Matthias HUSS, Daniel FARINOTTI, Philippe HUYBRECHTS

**General comments**

This paper produces with a lot of heavily calibrated model outputs some comparisons between local glacier-specific models and global-scale models. The main goal of the research questions is very unclear in relation to a real scientific outcome of the paper. This results from shortcomings in acknowledging and dealing with known processes and problems in this scientific field, ranging from highly parameterized models, over big uncertainties in climatic forcing reanalysis and future scenarios, to regional and global glacier mass balance estimation uncertainties. As a consequence, it is confusing to see some results presented in a publication of a journal addressing experts in glaciology, like 'The Cryosphere', with statements mentioned in the abstract section like '*Our findings thus suggest that when modelling small to medium-sized glaciers the emphasis should be on having a reliable reconstruction of the glacier geometry rather than focusing on a detailed representation of ice flow and mass balance processes'.* Here, the overgeneralized interpretation that in my opinion is insufficiently addressing these issues reads as if the authors' main goal is to explain finally the glaciological community that it is important to observe glacier geometry and volume. I think it is well known that the existing ice geometry and volume is fundamental for doing any long-term observations on glaciers and estimating their future state and we do not need a paper telling us this simple fact.

The "Models and Data" section is extremely vague and lacking important information, leaving it up to the reader  gather these crucial pieces of information in a large number of other papers; sometimes this also generates confusion in relation to **which version of a data/model** or **which type of calibration** is used. Some examples of missing information is mandatory to be presented within the manuscript and/or in a supplement:

- **Which are the time periods** (for each glacier) over which the models are calibrated against *in situ* measurements? **How many *in situ* measurements** (ll. 135-137) are actually used for each glacier? This could also be a collection of figures from previous papers, but in a modeling study like this it must be shown. It would for example show that accumulation measurements are sparse or even lacking for some glaciers. Accumulation rates are not only uncertain but also potentially evolving over long-term periods, is this somehow considered?

- Calibration of GloGEMFlow against the geodetic data of Hugonnet (ll. 150-154): the authors mention a calibration following Huss and Hock (2015), but they calibrate their parameters (precipitation gradient, degree-day factors and air temperatures) only against the bias. **Calibration of three or more parameters against a single number (the bias) is not very meaningful,** as it is always possible to adjust the model output until it matches the reference (geodetic) value. What about the spatial residuals (RMS)? The model could potentially be over-estimating mass balance by +999 m w.e. at certain locations, and under-estimating by -999 elsewhere! Thus, the **full results of calibration must be shown**, including residuals w.r.t. the single measurements. The modeled mass balance must be made available for download – that is the only way to see that the modeled mass balance is realistic.

The paper claims a main role of initial thickness in controlling the modeled glacier evolution, as opposed to the mass balance forcing and the choice of ice flow model complexity.

- ll. 373-375, *'The findings reveal that the initial ice thickness (volume) reconstruction strongly dictates the glacier's future evolution, with larger initial volumes resulting in more ice remaining by 2050 and the end of the 21$^{st}$ century (Figure 7).'.*

  However, the spreads in Fig. 3 look at least as large (and more temporally variable) as the spreads in Fig. 7 (excluding the red Millan line, which is reflecting a very large – probably wrong – ice thickness estimate). This would contradict the authors' statement: **the role of mass balance and ice dynamics appears to have the same order of magnitude as the initial thickness.** It would be useful to have a quantitative metric (e.g. $R^2$ or similar) of the relative importance of mass balance, ice dynamics and ice thickness (e.g., some summary number computed from Figure 10).

- ll. 461-462, *'One of the key outcomes of this study is the pronounced sensitivity of the projected future glacier volume to the selection of the ice thickness reconstruction employed for model initialisation (Figure 10).'*

   **Fig. 10 in the sense as it is interpreted in the paper.** Only for Grigoriev (purple diamonds in Fig. 10d) the difference is very high, and most of the difference likely comes from the (very wild) thickness estimate of Millan. Fig. 10 is showing a fundamental role of ice thickness (versus choice of mass balance and ice flow model). It would be useful to show an actual number summarizing the relative importance - the magnitudes in panels (a) and (b) appear to be similar to panel (d). Also, obviously the importance of initial ice thickness will be greatest for the near-future (as the glaciers need time to adapt) and decrease over time. In Fig. 10, for 2075 and 2100 the dots are not really higher in panel (d) than in panels (a-c), negating the main conclusion of the manuscript. Thus, initial ice thickness is maybe equally important as mass balance and ice flow, but not "mostly [important]" (l. 502).

- What about the importance of the future climate? Do the mentioned climate models (ll. 155-158) have any clue about local weather and its trends? The authors mention a "bias correction" procedure (l. 160) performed *'in order to align the output of the GCMs with observed climate data'.* How big of a change is that? How does GCM precipitation compare to measurements, is it any good? One could argue that in fact the input weather can be the largest uncertainty for the future, far exceeding that of thickness: especially in a regional model, the (virtually unknown) spatial variability of precipitation (on the ~1 km to ~100 km scale) could add vast amounts of variability in the glacier evolution, not captured by any of the used models. Thus **all the models would be similar (in part) because they all share a common lack of a key variability in the input.** Furthermore, the authors mention that the GCMs are aligned with observations (l.160) where it is assumed that the ERA5 dataset should represent the observations. However, calling ERA5 an observational dataset is incorrect and unsubstantiated as ERA5 has a lot of uncertainty and does not necessarily represent real situations, especially not in Central Asia (e.g. Zandler et al. 2019, Guo et al. 2021, Barandun and Pohl, 2023). Or are meteorological station data meant? This should be made clearer and which datasets the GCM runs are calibrated against. Furthermore, the reference as of why the specific CMIP6 climate model runs were selected seems inaccurate. The referenced paper seems to talk about hot biases but is not presenting a comparison of model runs that would then lead to the selection of the specific model runs. This would also need some justification/clarification or at least/best a comparison of these model runs so that the reader can get an impression of how different the model runs are. Optimally, this also includes the data after bias calibration.

**Impact of mass balance:**

The authors are comparing the results of mass balance forcing computed using the simple SEB of Oerlemans (2001) and using the degree-day approach of GloGEMFlow. **The Oerlemans SEB model** is calibrated against stakes and snow pits, but it **is still just a highly simplified melt model!** For example, the **effect of sublimation** – significant both as an energy sink and a mass balance component across dry Central Asia – **is ignored by both models**. Using a full energy balance model (like the EBFM or COSIPY) as "local" model would be more meaningful, as it would really show the impact of including vs excluding the actual mass balance processes. Also, for a better understanding of the model comparison, **it would be important to at least provide a list of the processes which are included in each model** – what about the future evolution of albedo and of debris cover of the glaciers? I suspect it might be included in the simple, regional GloGEMFlow but not in the local Oerlemans SEB model?

**Impact of ice flow:**

- Surely the *'3-dimensional higher-order thermomechanical ice-flow model (3D HO-model)'* does not have only two parameters *('the enhancement factor and the basal sliding parameter'*, l. 183)? The manuscript shows that it is possible to run the 3D HO-model and obtain a result which is similar to that of the simpler GloGEMFlow; but I expect **it is also possible to obtain a very different result while still using reasonable parameter values:** what about the impact on future glacier evolution of the other parameters in the model? **A sensitivity study is expected here**, or, if already performed in Van Tricht and Huybrechts (2023, still a preprint!), its main results must be summarized here. Unless the authors can claim that all other parameters have a negligible impact on future glacier evolution, a reasonable uncertainty estimate contributed by the other parameters must also

be considered and is potentially significant (e.g. within Fig. 10).

- Can any of the used models simulate dynamic glacier instabilities? **A large fraction of glaciers in Central Asia are known to be of surge type**, while the 6 selected glaciers (to the best of my knowledge) are currently not. The accuracy of a model on regional scale will certainly be affected by the presence of surging glaciers, which can radically alter the hypsography, topography and surface morphology of a glacier, leading to large variations in mass balance not directly linked to the climate. This would limit the validity of the study's results to small, stable glaciers. It is also important to consider the temporal evolution of such instabilities – for instance:

  - Increasingly prevalent glacier instabilities, possibly linked to a large-scale cold-polythermal-temperate transition.

  - Formerly unstable glaciers, made stable (no longer surge-type) by the changed morphology following retreat.

The main author uses mainly self-citations of his studies, which he probably did during his PhD. However, all the older studies from well-known Central Asian colleagues, published in the past like the studies of Glazirin, Aizen, Dyurgerov, Dikih, etc. are nearly not mentioned. Even if the scientist is not able to read old Russian literature, it is necessary to find a way to include the findings of this literature for many aspects of glacier evolution in the region were already assessed and reported.

Overall, the outlined points render this paper very vague, and the reader is left guessing what exactly is being compared to what.

**Specific comments**

- ll. 144, This citation is an Egusphere abstract, where the 'comprehensive information' is not findable as the information in the abstract is very limited.

- ll. 149-150, ERA-5 reanalysis in Central Asia should not be taken as a gold standard as recent studies have shown (e.g. Zandler et al. 2019, Guo et al. 2021, Barandun and Pohl, 2023, ).

- ll. Table 1: Description of ice thickness datasets. Why are the measurements of the ice volume not shown in Table 1? Please give some more details about GPR measurements and how they have been inter-and extrapolated.

- ll. Table 1: The paper uses the results from Millan et al., 2022. Why do the authors take this paper as a comparison, if they obviously do **not agree** with the ice thickness results presented in Millan et al. 2022. The authors already have the results from Farinotti et al. 2019a and all the results from Model 1 to Model 4.

- ll. 228-229, The observation on Kara-Batkak suggest a shorter response time and faster approach to equilibrium. Please show the calculated response time for each glacier.

- ll. 232, According to the interpretation of your figure (Fig. 3), this is not true. See Ashu-Tor and other glaciers in Figure 3. You mention that regional mass balance and local mass balance are the same. This is in my view not correct and not supported in the figure, e.g. certain glaciers like Ashu-Tor in Figure 3, where the difference of the volume at 0.2 is around 20 years.

- ll. 233-235, If you compare a local to a regional mass balance model, then the focus should still be on the differences of the individual glaciers and not on an aggregated ice volume. This is because the differences at individual glaciers can actually reveal similar or different behaviour, whereas the aggregated ice volume is affected anyways from the calibration.

- ll. 298-301, Give more details exactly for this calibration process?

- ll.303-305, This sentence is not conclusive. What means *'…peculiar local characteristics that are difficult to capture'.* Please be more specific.

- ll. 329, Surface elevation from SRTM is well known to have large uncertainties because of penetration of the radar waves into the firn area.

- ll. 371-375, This result (*'the initial ice thickness (volume) reconstruction strongly dictates the glacier's future evolution'*) is expected when comparing glaciers with significantly different ice volumes regarding their disappearance time. But glacier evolution is more than disappearance time and this statement seems unsubstantiated with the lack of information and considerations about process descriptions and uncertainties in climate forcing as outlined in the previous comments.

- ll. 427-428, What are the extrapolation techniques, which are mentioned in the caption of Figure 9?

- ll. 438-440, This statement clearly shows that not only geodetic mass balance will deliver the best results, but instead, also local data is fundamental to understand the local differences which are normally **not presented in models like GloGEM**. Inclusion of special processes such as long-term change of temperature conditions within glaciers, changes in runoff based on changes in temperature or mass balance, changes in flow regimes, sublimation effects, long-term changes in pore conditions of accumulation areas are often leading to changes, which are not covered at all in these models but are significantly changing the behaviour of the reaction of a glacier. The authors should be more careful with their statements. Simplification is sometimes ok but here the authors oversimplify.

- ll. 443-444, This is still only an assumption if you calibrate your parameters in a way that both models fit. However, this is not showing all the uncertainties if you would change your parameters particularly in the complex model.

- ll. 461-467, This is commonly accepted and taught at universities. There is no need to have a paper telling specialists in this research field that existing glacier volume is fundamental for studying glacier change.

- ll. 480-481, The author writes: '*highlights that a global-scale flowline model is capable of accurately simulating glacier dynamics and evolution*'. This quite generic claim is just wrong. "[...] accurately simulating glacier dynamics [...]" would include the simulation of observed processes such as seasonal velocity changes and glacier surges.

References

Barandun, M. and Pohl, E., 2023. Central Asia's spatiotemporal glacier response ambiguity due to data inconsistencies and regional simplifications. The Cryosphere, 17(3): 1343-1371.

Guo, H., Bao, A., Chen, T., Zheng, G., Wang, Y., Jiang, L. and De Maeyer, P., 2021. Assessment of CMIP6 in simulating precipitation over arid Central Asia. Atmospheric Research, 252: 105451.

Zandler, H., Haag, I. and Samimi, C., 2019. Evaluation needs and temporal performance differences of gridded precipitation products in peripheral mountain regions. Scientific Reports, 9(1): 15118.